# Drill-Down Analysis of LLM Hallucination Patterns in Text-to-SQL

## Abstract

Despite impressive benchmark scores, Large Language Models (LLMs) can still produce flawed and incorrect responses for Text-to-SQL tasks. While prior work has decomposed complex SQL queries in an attempt to improve LLM benchmark performance, few have systematically analyzed hallucination propagation patterns within these decomposed structures. We present a drill-down evaluation framework that decomposes complex SQL queries and questions from the BIRD-mini dataset Li et al. (2023), allowing for a fine-grained analysis of hallucination propagation. Through our analysis, we report three key findings: (1) Recurrent Hallucinations: Many hallucinations persistently propagate from early, structurally simple sub-queries through to final steps, indicating systematic misalignment. (2) Final-Step Emergence: Fewer, but specific hallucination types emerge in the final step, suggesting a distinct failure mode tied to query complexity. (3) History Amplifies Recurrence: While contextual information between sub-queries can help to reduce the frequency of emergent hallucinations, it consequently increases the recurrence of early-stage hallucinations. This framework establishes a methodology to better understand LLM weaknesses and failure modes for Text-to-SQL systems.

## 1 Introduction

Recent advances in Text-to-SQL techniques represents a significant leap forward in human-computer interaction, promising users the ability to query complex databases using everyday conversational language instead of structured query syntax. At the heart of this transformative technology are Large Language Models (LLMs), which have demonstrated remarkable proficiency in this task Li et al. (2024a); Chen et al. (2024); Hong et al. (2024). By leveraging their vast pre-training on diverse text and code corpora, LLMs can grasp the semantic intent behind a natural language question, understand the underlying database schema, and generate an executable SQL query to retrieve the correct information. This capability is poised to democratize data access, empowering non-technical stakeholders to directly interact with data and derive insights without the need for specialized programming skills, thereby accelerating the pace of data-driven decision-making. However, hallucinations introduced by the LLM remain a consistent and persistent issue in all Text-to-SQL via LLM pipelines. These hallucinations are a notorious problem in LLMs and refer to instances where they generate content that is irrelevant, erroneous, or inconsistent with the user's requests Huang et al. (2023); Qu et al. (2024); Zhang et al. (2024). While researchers are aware of hallucinations, interpreting, explaining, and preventing them remains an open area of research.

Crucially, in a text-to-SQL task, a hallucination isn't just a factual error but a functional failure that represents a key challenge for AI alignment. An incorrect query could lead to the wrong business decisions, faulty reports, or even data corruption if the system is designed to execute the queries without human oversight. Ensuring the LLM produces safe, reliable, correct and intention-aligned SQL is a fundamental alignment challenge. Furthermore, users will quickly lose trust in a system that consistently produces queries that fail to execute or return incorrect data. An aligned system is one that a user can trust to perform its task reliably. Hallucinations erode this trust, which is a clear symptom of misalignment. While a human can often catch these errors, a truly aligned system should minimize the need for a human to constantly debug its output. The goal of text-to-SQL is to empower non-technical users, but hallucinations make this difficult and require a level of technical expertise to correct.

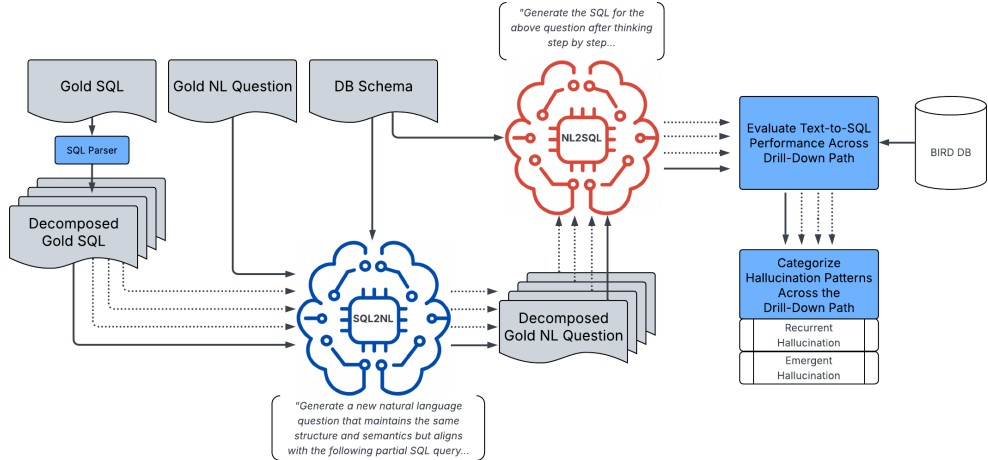

Figure 1: Drill-down framework; Decompose the BIRD-mini dataset into progressive sub-queries and sub-questions and evaluate their hallucination patterns.

We present a drill-down hallucination framework and analysis in the Text-to-SQL domain. First, we decompose the SQL queries into sub-queries which generates a custom drill-down dataset from an existing Text-to-SQL dataset (Fig. 1). Second, we create an automated pipeline for annotating a LLMs hallucinations, with a fine-grained taxonomy which builds temporal abstractions on top of hallucination categories identified by prior research Qu et al. (2024). This novel analysis enables a deeper investigation into how different types of hallucinations evolve across multi-step generation paths for Text-to-SQL. Using these annotated results we analyze for *recurrent hallucinations*, where the same erroneous instances persist from earlier steps into final outputs, and *emergent hallucinations* that appear for the first time in final reasoning steps despite having no prior instances. Since SQL composition gradually increases in complexity, by defining recurrent and emergent hallucinations is, we can determine whether hallucinations originate from earlier stage conditions or from the model's difficulty coordinating the full complexity of the final stage.

Our results reveal interesting insights into hallucination patterns and failure mechanisms that are consistent across six modern LLMs. Understanding and addressing these patterns would provide a deeper understanding of these models and provide a path to better alignment. Overall, this paper evaluates six modern LLMs, two from Anthropic Anthropic (2024; 2025) and four from OpenAI OpenAI (2023; 2024; 2025) in the Text-to-SQL domain, analyzing their hallucinations to better understand the weaknesses of these models. Briefly, the contributions of this paper can be summarized as follows:

1. We leverage the decomposable nature of SQL queries to create a drill-down analysis pipeline that provides an insight into LLM hallucinations when used in text-to-SQL pipelines.

2. Our experiments uncover two distinct temporal (w.r.t to the sub-query step) failure patterns, described as recurrent and emergent hallucinations.

3. We evaluate six closed-source frontier models (Claude and GPT variants) and show that hallucination patterns are consistent across architectures and vendors.

## 2 RELATED WORK

Early Text-to-SQL systems almost always adopted a sequence-to-sequence framework in which both the natural-language question and the target database schema were jointly encoded by neural models. Early efforts relied on recurrent architectures for this encoding Dong & Lapata (2016); Jia & Liang (2016), before moving toward graph neural networks that explicitly model schema structure Bastings et al. (2018); Bogin et al. (2019), and, eventually, to pre-trained transformer encoders Yin et al. (2020); Yu et al. (2021). More recently, LLMs have become a dominant paradigm due to their

strong generalization ability, few-shot learning capacity Brown et al. (2020), in-context reasoning Xie et al. (2021), and chain-of-thought prompting capabilities Wei et al. (2022). These capabilities allow LLMs to generate SQL queries from natural language with little to no task-specific fine-tuning. Although this transition has led to notable performance gains on standard benchmarks, it has also introduced new challenges, one being hallucinations.

Recent papers have introduced new and unique approaches to improve performance and better align the Text-to-SQL system with the given task. CHASE-SQL Pourreza et al. (2024) represents a recent methodology that uses a divide-and-conquer strategy to decompose complex problems into sub-components, addressing each component separately before synthesizing the results into a final solution Pourreza et al. (2024). This technique shows impressive performance improvements on the BIRD benchmark's Li et al. (2023) execution accuracy (EX) metric. Inspired from this framework, we decompose the complete BIRD-mini dataset Li et al. (2023), breaking it down into sequential sub-components. However, our approach is different from existing research that leverages decomposition primarily as a step for benchmark optimization. Instead, we conduct a systematic analysis of hallucination behaviors and patterns within these decomposed structures.

## 3 DATASET

**BIRD-mini** We conduct our experiments on the PostgreSQL The PostgreSQL Global Development Group (1996–2025) BIRD-mini dataset, a smaller version of the full BIRD benchmark specifically designed to capture the complexity and diversity of BIRD while keeping the experiments feasible for resource-constrained researchers. Recent work has used BIRD-mini for multi-turn Text-to-SQL interaction Meng et al. (2025)(CIKM 2025) and industrial NL2SQL agents Jeon et al. (2025)(NeurIPS TRL workshop). Applying our hallucination annotation and decomposition framework directly to the full BIRD-dev split would more than double its size, requiring tens of thousands of additional sub-questions and queries, leading to a prohibitive API cost across six LLMs. Instead of arbitrarily sub-sampling BIRD-dev, we expand BIRD-mini, which has already been curated by the BIRD authors as a representative, high-quality, and cost-effective subset for Text-to-SQL development Li et al. (2023). We expand BIRD-mini to 1383 instances and evaluate this expanded dataset across six modern LLMs. This expansion reflects the maximum possible decomposition where each sub-query remains executable, yielding 1383 systematic question–query pairs.

**Our drill-down pipeline can be applied to the full BIRD dataset or other benchmarks** with minor configuration changes. In this work, we use BIRD-mini purely for cost reasons; the framework itself scales to larger benchmarks. We have also validated the framework's generality by applying it to a subset of Spider, confirming that it functions as expected beyond BIRD-mini (Appendix A).

## 4 HALLUCINATION TAXONOMY

**Schema-Based and Logic-Based** For a more accurate categorization of these hallucinations, we adopt the taxonomy featured in Qu et al. (2024). which categorizes hallucinations into two main categories, schema-based and logic-based. *Schema-based hallucinations* reflect misunderstandings of the database structure itself, using incorrect tables/columns or unnecessarily attributes. *Logic-based hallucinations* involve errors in how the query is constructed, unnecessary joins, clause abuses, or incorrect math. We describe these hallucination categories in more detail in Appendix B, Table 1.

**Recurrent and Emergent Hallucinations** Beyond the taxonomy, we will additionally define two more hallucination behavior types that capture distinct patterns. The first is *recurrent hallucinations*, which we define as a hallucination that occurs somewhere in the drill-down path and reappears in the final step. These errors demonstrate persistence across multiple steps of the drill-down path, suggesting a more fundamental misunderstanding. The second is *emergent hallucinations*, which we define as a hallucination that only occurs in the final step of the drill-down path. These errors appear to be triggered specifically by the increased complexity and integration requirements of the complete problem. These categories can be viewed as temporal abstractions, with respect to the sub-query steps, over the hallucination categories identified in Qu et al. (2024).

**Failure Mechanisms**   For this paper, we interpret these behavioral distinctions as two different failure mechanisms occurring with these LLMs. Recurrent hallucinations manifest not only when confronted with the original complex BIRD-mini question, but they also persist in identical ways when presented with the decomposed versions of the same problem. Emergent hallucinations, conversely, capture a failure mode that occurs uniquely with the full complexity of the question and query. These failures suggest that models can successfully navigate some components of a complex problem but fail when required to synthesize the final complexity of the original problem.

## 5   METHODOLOGY

This section will outline the framework we used to perform our drill-down analysis of hallucination patterns, consisting of three primary components:

1. Decompose the BIRD-mini dataset into progressive sub-questions and sub-queries.

2. Perform drill-down evaluation on multiple LLMs.

3. Categorize and describe the hallucination patterns (Fig. 1).

Follow Algorithm 1 for each step of our framework.

---

**Algorithm 1** Drill-Down Hallucination Analysis on BIRD-Mini

---

**Input:** Original dataset $\mathcal{D} = \{(q_i, s_i)\}_{i=1}^{N}$; schema $\mathcal{S}$
**Output:** Annotated failure set $\mathcal{H}$ with hallucination categories
**Initialize:** $\mathcal{P} \leftarrow \emptyset, \mathcal{H} \leftarrow \emptyset$
**foreach** $(q_i, s_i) \in \mathcal{D}$ **do**
    $\{s_i^j\}_{j=1}^{K_i} \leftarrow \text{DECOMPOSE}(s_i)$
    **for** $j = 1$ **to** $K_i$ **do**
        $q_i^j \leftarrow \text{LLM\_REWORD}(q_i, s_i^j, \mathcal{S})$ $\mathcal{P} \leftarrow \mathcal{P} \cup \{(q_i^j, s_i^j, s_i)\}$
**foreach** $(q, s^*, s_{\text{full}}) \in \mathcal{P}$ **do**
    $\hat{s} \leftarrow \text{LLM\_GENERATESQL}(q, \mathcal{S}, s_{\text{full}})$
    **if** $\text{EXECACCURACY}(\hat{s}) = 0$ **then**
        $\mathcal{C} \leftarrow \text{CATEGORIZEFAILURE}(\hat{s}, s^*, \mathcal{S})$   $\mathcal{H} \leftarrow \mathcal{H} \cup \{(q, \hat{s}, s^*, \mathcal{C})\}$
**return** $\mathcal{H}$

---

### 5.1   DECOMPOSE AND GENERATE DRILL-DOWN DATASET

**Progressive Sub-Query Generation**   The proposed framework begins by decomposing each query from the BIRD-mini benchmark into multiple progressive queries, using an SQL parser Albrecht (2024). By parsing progressively from **select** through **where** and subsequent **and** conditions, we ensure that each sub-query in the drill-down path represents an executable SQL query. Follow Fig. 1 for an example.

**Sub-Question Generation**   We additionally pair each of these sub-queries with a sub-question that captures the contents of the sub-query in natural language (NL). To ensure the reliability of our expanded benchmark, we adopt an asymmetric design choice: all sub-queries are generated deterministically via sqlparse, while sub-questions are produced by LLMs (GPT-4o-mini) provided with the BIRD database schema, original question, and our generated sub-queries. We additionally regenerate the original question with the same method to maintain alignment with the generated sub-questions. This choice follows recent evidence that formal language → natural language (SQL-to-NL) is consistently more reliable than the reverse natural language → formal language (NL-to-SQL).

For example, Evaluating NL-to-SQL via SQL-to-NL shows that SQL-to-NL achieves stronger Pass@K performance on Spider and produces paraphrases with higher semantic fidelity and fewer schema-alignment errors than NL-to-SQL Li et al. (2025). These findings support our claim that LLM-generated sub-questions faithfully capture the meaning of their corresponding SQL sub-queries, with lower risk of hallucination compared to direct NL-to-SQL generation. Nevertheless,

this step may introduce subtle artifacts that could influence the hallucination analysis in Section 6, and should therefore be considered when interpreting the results.

## 5.2 DRILL-DOWN AND ANNOTATE HALLUCINATION PATTERNS

Following this process, we construct an incremental sequence of questions and queries that gradually increases in complexity. We transform and expand the original BIRD-mini dataset into a drill-down dataset which enables us to pinpoint precisely where hallucinations emerge within these incremental pathways and determine whether these errors propagate to the final stage. We categorize and annotate these hallucination types and behaviors. **A full description of the heuristics and rules we used to annotate the hallucination types is shown in Appendix D.**

## 6 EXPERIMENT

We systematically evaluated six LLMs, Claude-3.5-sonnet and Claude-3.7-sonnet from Anthropic Anthropic (2024; 2025), and GPT-4-turbo, GPT-4o-mini, GPT-4.1-mini, and GPT-4-nano from OpenAI OpenAI (2023; 2024; 2025) on our BIRD-mini drill-down dataset for the Text-to-SQL task using the default prompt provided by BIRD (Appendix E) Li et al. (2024b). To uncover where and how hallucinations arise, we perform a structural comparison between predicted SQL, ground-truth SQL, and the database schema at each step of a progressive question path. Each hallucination is categorized and annotated through this multistep decomposition.

The experiments are designed to address the following research questions:

**Research Question 1** Do hallucinations in Text-to-SQL generation primarily originate from the complexity of the original question, and/or do they instead emerge earlier due to misunderstandings in simpler steps? This question is inspired by recent research from Qu et al. (2024) positing that hallucinations often arise when models treat decomposed sub-tasks as entirely novel and must generalize from scratch, rather than leveraging prior experience.

**Research Question 2** What hallucination types emerge uniquely at the final stages of Text-to-SQL generation, and how are these failures correlated with query complexity? We ask this question because we wish to better understand what hallucination types are emergent and which are recurrent. More specifically, for the emergent hallucinations we observe, can we correlate this type with the query complexity at this final step?

**Research Question 3** How does access to contextual history from the drill-down path during Text-to-SQL generation affect the frequency and severity of recurrent versus emergent hallucinations? What type of role does context history play for LLMs when processing across a drill-down path, where the context increases along with the complexity?

## 6.1 EVALUATION METRICS

We adopt the problem formulation from Qu et al. (2024). Given a natural language question $\mathcal{Q} = \{q_1, \ldots, q_{|\mathcal{Q}|}\}$ and its associated database schema $\mathcal{D} = \langle \mathcal{C}, \mathcal{T} \rangle$, where $\mathcal{C} = \{c_1, \ldots, c_{|\mathcal{C}|}\}$ and $\mathcal{T} = \{t_1, \ldots, t_{|\mathcal{T}|}\}$ represent the sets of column and table names respectively, the goal of the text-to-SQL task is to generate a valid SQL query $y$ that faithfully reflects the intent encoded in $\mathcal{Q}$.

**Execution Accuracy (EX)** We evaluate baseline model performance using two main metrics, the first being *Execution Accuracy (EX)* Li et al. (2024a), which measures whether a predicted SQL query $\hat{y}$ yields the same execution result as the ground truth query $y^*$ when both are executed on the same database instance. Formally, let $\texttt{Exec}(y, \mathcal{D})$ denote the result of executing query $y$ on database $\mathcal{D}$. Then, the EX score for a single example is defined as:

$$\text{EX}(\hat{y}, y^*) = \begin{cases} 1 & \text{if } \texttt{Exec}(\hat{y}, \mathcal{D}) = \texttt{Exec}(y^*, \mathcal{D}) \\ 0 & \text{otherwise} \end{cases}$$

The overall EX score across a dataset of $N$ examples is computed as the average:

$$\text{EX}_{\text{avg}} = \frac{1}{N} \sum_{i=1}^{N} \text{EX}(\hat{y}^{(i)}, y^{*(i)})$$

**Soft-F1 Score**    The second metric we use is the *Soft-F1 Score* Li et al. (2024a). Unlike Execution Accuracy, which is binary and requires an exact match in result sets, Soft-F1 provides a graded assessment by measuring partial overlaps between the execution results of the predicted and ground truth SQL queries. Let $\hat{T} = \texttt{Exec}(\hat{y}, \mathcal{D})$ and $T^* = \texttt{Exec}(y^*, \mathcal{D})$ be the predicted and ground truth result tables, respectively. At the tuple level, treating each tuple as a set of values, define;

- True Positives (TP): $\hat{T}$ and $T^*$
- False Positives (FP): $\hat{T}$ but not in $T^*$
- False Negatives (FN): $T^*$ but not in $\hat{T}$

The Soft-F1 score is then computed as:

$$\text{Soft-F1} = \frac{2 \cdot \text{TP}}{2 \cdot \text{TP} + \text{FP} + \text{FN}}$$

## 6.2 RESULTS

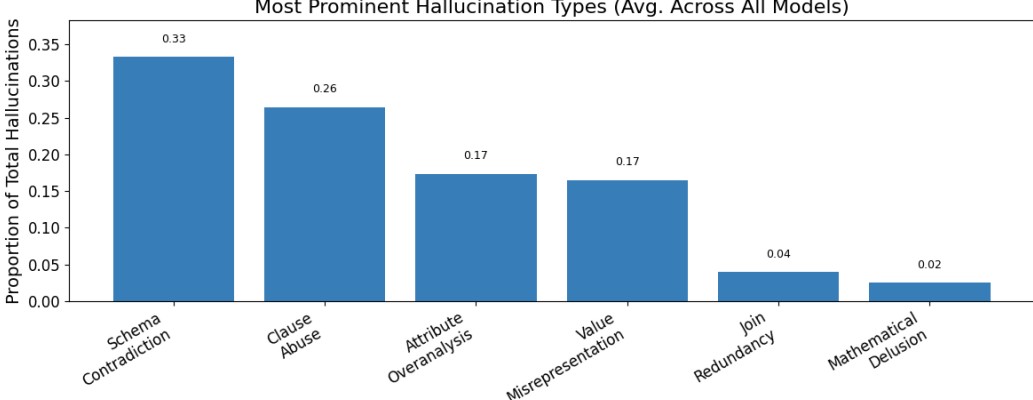

Figure 2: Distribution of hallucination types across all experiments. Schema contradiction and clause abuse emerge as the dominant categories, indicating that models frequently misinterpret schema structure or over-apply SQL clauses even in decomposed forms.

To validate our setup, we first report baseline performance on BIRD-mini, showing close alignment with previously reported scores Li et al. (2024a), as shown in Appendix C, Table 3. Additionally, Fig. 2 displays the most prominent hallucination types across all of the experiments conducted (average across all models).

$P(\textbf{In Final Step} \mid \textbf{Occurs in Earlier Steps})$    Fig. 3 presents the conditional probabilities of hallucinations occurring in the final step (original BIRD-mini question) given that the identical hallucination type manifested earlier in the drill-down path, expressed as $P(\text{In Final Step} \mid \text{Occurs in Earlier Steps})$. The results reveal that hallucinations are not exclusively confined to the final, most complex step, but rather demonstrate recurrence patterns throughout earlier stages of the progressive path. Notably, while Schema-Based: Schema Contradiction and Logic-Based: Clause Abuse represent the two most common hallucination types in our results Fig. 2), they seem to exhibit different failure mechanisms. Most hallucination types exhibit relatively high recurrence probabilities, with the exception of Logic-Based: Clause Abuse, see (Fig. 3). The persistence of these errors

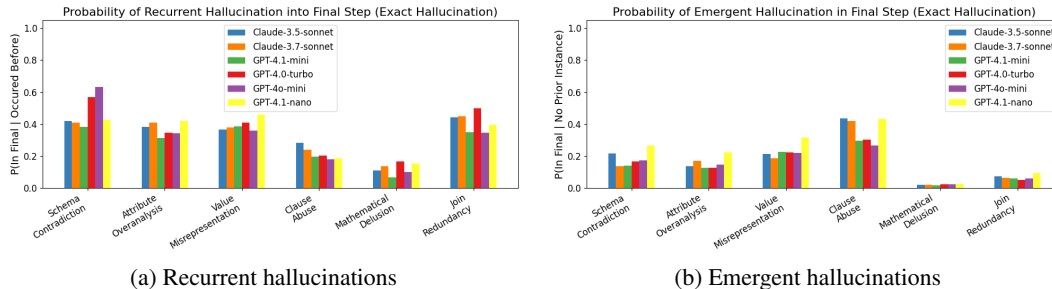

(a) Recurrent hallucinations        (b) Emergent hallucinations

Figure 3: Probability of recurrent (a) and emergent (b) hallucinations across categories (exact same hallucination). Recurrent errors show high persistence once introduced (sometimes >50%), while emergent errors are rarer, with clause abuse being the main exception. This highlights distinct failure mechanisms between persistence and final-step emergence.

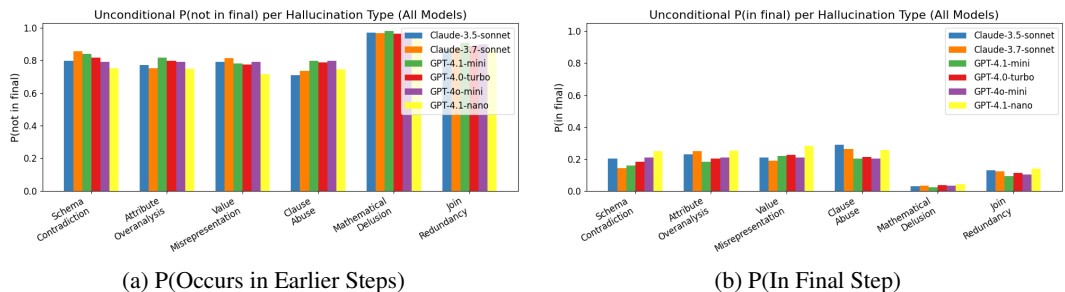

(a) P(Occurs in Earlier Steps)        (b) P(In Final Step)

Figure 4: Unconditional probabilities, P(Occurs in Earlier Steps) (a) and P(In Final Step) (b) by hallucination types. For (a) We see a more even distribution across all categories with Mathematical Delusion and Join Redundancy having the highest probabilities. For (b) Mathematical Delusion and Join Redundancy have the lowest probability while Clause Abuse appears to have one of the highest probability of being in the final step.

across multiple stages, including the initial steps of the path, indicates fundamental misalignment issues where LLMs struggle with a task even in their most decomposed forms.

$P(\textbf{In Final Step} \mid \textbf{Does Not Occur Earlier})$     Conversely, Fig. 3, shows similar probabilities but for hallucinations that occur in the final step where the exact same hallucination does not occur anywhere in the drill-down path, $P(\text{In Final Step} \mid \text{Does Not Occur Earlier})$, we observe a distinctly different pattern. Most hallucination types exhibit considerably lower emergence probabilities compared to their recurrence rates, except for Logic-Based: Clause Abuse, which has a higher probability of emergence compared to recurrence. The lower probabilities suggest that most hallucination types are more likely to propagate from earlier steps, with the outlier being Clause Abuses.

$P(\textbf{In Final Step})$ and $P(\textbf{Occurs in Earlier Steps})$     Fig. 4 shows the unconditional probabilities of hallucination types occurring; in the final step, or in earlier steps. These results seem to follow some of the patterns observed for the conditional probabilities. We see that the results for P(In Final Step) displays a similar distribution to the emergent hallucinations table. However, Clause Abuse is much less pronounced in this unconditional table. Additionally, the results for P(Occurs in Earlier Steps) shows much higher and uniform distributions for all hallucination types with Mathematical Delusion and Join Redundancy having the highest probabilities (>80%). Similar to our conditional probabilities table $P(\text{In Final Step} \mid \text{Occurs in Earlier Steps})$ we see higher probabilities for the unconditional P(Occurs in Earlier Steps) table. These results make sense intuitively as we see a much higher density for our P(Occurs in Earlier Steps) vs P(In Final Step). P(Occurs in Earlier Steps) allows for more chances of hallucinations of the same type during the multiple steps of the drill-down path, while P(In Final Step) is restricted to the final step.

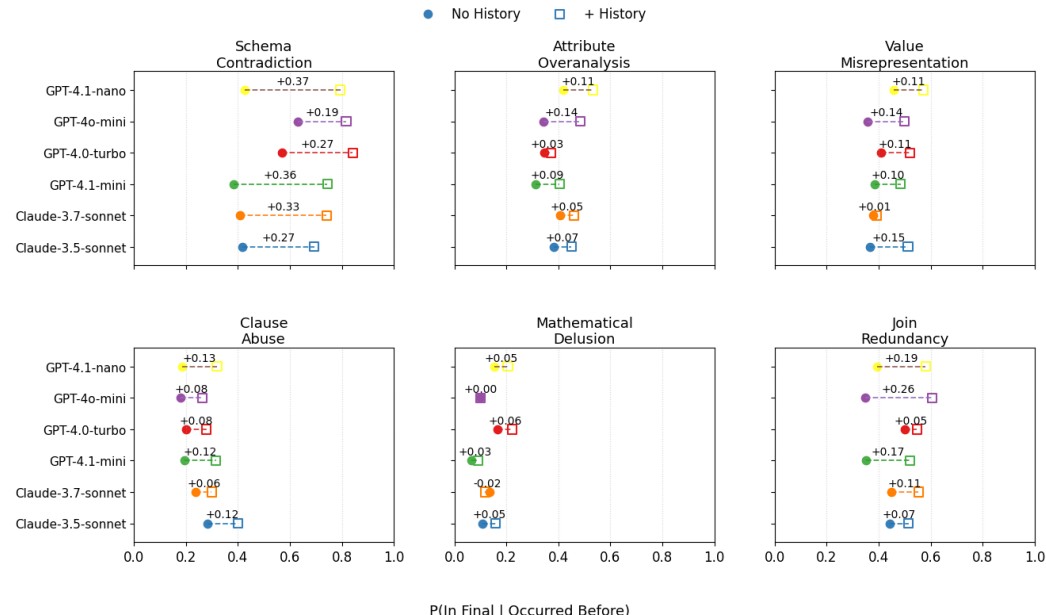

Figure 5: Probability of recurrent hallucinations with and without history/context. Providing history consistently increases recurrence rates across models, showing that context can inadvertently reinforce early-stage errors rather than correcting them.

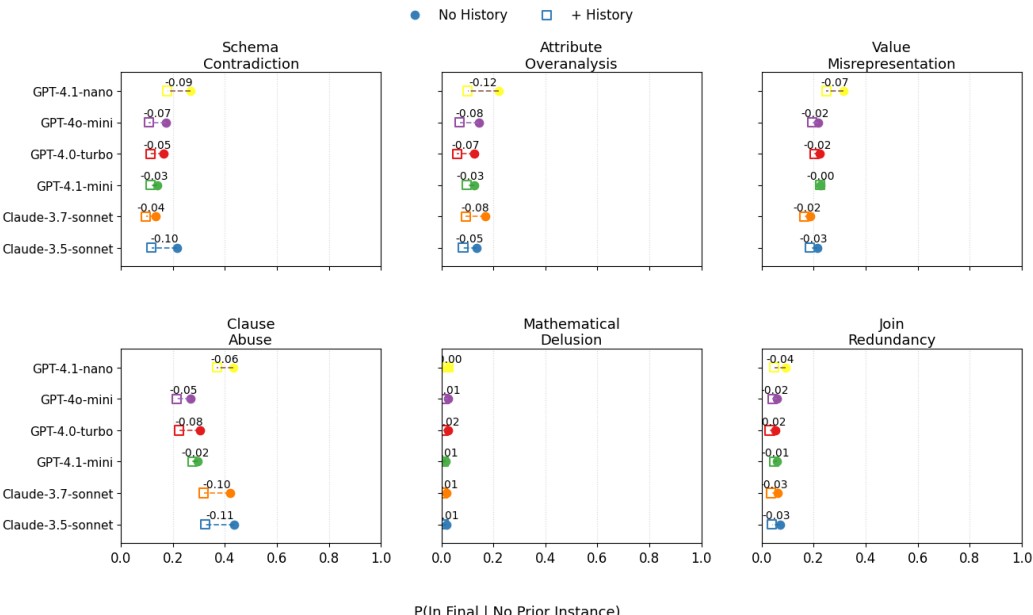

Figure 6: Probability of emergent hallucinations with and without history/context. In contrast to recurrent patterns, history reduces emergence rates, suggesting a protective effect against new errors but at the cost of amplifying persistent ones.

**LLM History Attention** Furthermore, examination of the results comparing history attention to the progressive path versus no attention reveals an interesting duality in hallucination behavior patterns. When models maintain access to conversational history throughout the progressive path, we observe a significant increase in the probability of recurrent hallucinations across all tested models compared to the context-free condition (Fig. 5). This suggests that contextual memory can inadvertently reinforce hallucination patterns established in earlier steps. In contrast, the presence of history

attention demonstrates a more protective effect against emergent hallucinations, reducing the probability of occurrence (Fig. 6). These results point towards a potential trade-off between emergent hallucination protection and recurrent hallucination amplification.

**Quantitative Analysis**    The strongest model (GPT-4.1-Mini) achieves 42.8% EX and 46.5% F1, while the weakest (GPT-4.1-Nano) records 30.0% EX and 33.49% F1. We also notice that performance drops rapidly with query complexity. Even with easy questions, the best EX reaches 59.46%, but drops to only 21.57% on challenging queries. We also find that adding contextual history decreases EX by ∼1.1 points and F1 by ∼0.6 points. For example, Claude-3.7-Sonnet improves slightly (38.8 → 39.8 EX), whereas GPT-4.1-Mini drops (42.8 → 40.0 EX). Figures 5–6 further demonstrates how recurrent errors seem to dominate and once they occur, they reappear in the final step with probabilities exceeding 50% for schema contradictions. Whereas, emergent hallucinations are less frequent for most hallucination types, excluding clause abuses. Finally, history impacts these distributions, raising recurrence anywhere from ∼2–37% across categories while reducing emergence by ∼2–11%.

# 7    THEORETICAL ANALYSIS

In this section, we interpret the patterns identified in our experiments to help explain some of the mechanisms driving hallucination in Text-to-SQL systems. The goal of this section is to connect the observed results to broader theoretical principles regarding Text-to-SQL failure formation and propagation. We now answer the three research questions introduced in Section 6 by examining how hallucinations originate, evolve, and respond to contextual history across the progressive query path.

**Research Question 1**    Hallucinations often originate in early sub-queries rather than only at the final complex step. High values of $P(\text{In Final Step} \mid \text{Occurs in Earlier Steps})$ for many types indicate that once a failure appears in a simple sub-query, it is likely to persist. This supports the view that some failures are driven by misunderstandings already present at low complexity.

**Research Question 2**    Emergent hallucinations are dominated by Clause Abuse, which appears primarily in the final query despite being absent in earlier steps. This suggests that certain logic-based errors arise specifically when the model must decide which global clauses are needed for the full problem, rather than from earlier misinterpretations. Additionally, the final step introduces the greatest number of global clauses, making it the stage where clause-related hallucinations are most likely to appear.

**Research Question 3**    Contextual history has a dual effect: it amplifies recurrent hallucinations by repeatedly exposing the model to earlier erroneous patterns, while simultaneously reducing emergent hallucinations by providing more information about prior reasoning and partial successes. This duality implies that history is beneficial for stabilizing correct patterns but can also stabilize incorrect ones.

## 7.1    IMPLICATIONS FOR MITIGATION

Now, we will describe three implications for mitigation of hallucinations in Text-to-SQL based on our research questions and findings. These implications are not prescriptive but rather conceptual guides resulting from our experiments on how hallucinations arise, propagate, and interact with system design choices. Each reflects a distinct structural property of hallucination behavior we observed.

**Early-step verification:**    This is motivated by the fact that both our conditional and unconditional probabilities showed a much higher density of hallucination when looking at the progressive paths before the final step. Because many failures originate in simpler sub-queries and then propagate, systems should focus on verifying or repairing early steps before allowing the model to build the full query.

**History usage as a design choice:** Based on our results it seems that contextual history is not uniformly beneficial. For tasks where recurrent schema failures dominate, truncated or selectively filtered history may be preferable to full history. Conversely, for tasks where emergent clause-related errors are prevalent, providing history may reduce final-step failures.

**Schema alignment as a priority:** The dominance and recurrence of Schema Contradiction reinforce the importance of pre-generation schema alignment, as emphasized by Qu et al. (2024). Our temporal analysis shows that misaligned schema usage tends to persist even after decomposition.

We further emphasize that these mitigation implications are exploratory insights based on our experiments, not finalized techniques.

## 8 EXPANDING ON PRIOR WORK

**"Before Generation, Align it!"** Qu et al. emphasizes the importance of *pre-generation alignment* between natural language and schema to mitigate schema-related hallucinations Qu et al. (2024). Our results support the claim that schema contradiction is one of the most prominent type of hallucinations in the Text-to-SQL domain. Furthermore, we have consistent results showing how recurrent schema hallucinations frequently persist into the final steps for all models tested (Figure 5).

**"A Study of In-Context-Learning-Based Text-to-SQL Errors"** Shen et al. present a taxonomy of 29 error types in in-context-learning (ICL) text-to-SQL Shen et al. (2025). This study quantifies overall error prevalence and repair challenges, we examine how these error types behave over the course of multi-step drill-down generation. By introducing *recurrent* and *emergent* hallucinations, we provide a new temporal perspective that extends beyond a static categorization.

## 9 CONCLUSION

Large language models (LLMs) currently demonstrate excellent capabilities in a variety of tasks, including text-to-SQL. However, hallucinations generated from the outputs of these models pose serious challenges for interpretability, alignment, and overall adoption into text-to-SQL systems. In this paper, we conduct a drill-down analysis to trace where in progressive query paths hallucinations arise. Our findings align with recent research that hallucinations can arise when models misinterpret the decomposed stages of a task as entirely new challenges Qu et al. (2024). However, we also find that it is common for hallucinations to reappear from earlier, and much simpler, steps into the final complex query. Finally, we report an inverse relationship with emergent and recurrent hallucinations when context to the drill-down path is provided to the LLM. We see a more protective behavior for emergent hallucinations but inversely an amplifying effect for recurrent hallucinations.

Our experiments reveal interesting nuances of LLM Hallucinations in the Text-to-SQL domain, providing researchers with a deeper insight into how these models are failing. Our hope is that these findings can help guide future work to improve Text-to-SQL systems. Ultimately, we aim for these insights to spur new research and alignment strategies that treat hallucination not as an unpredictable artifact, but as a diagnosable and preventable failure mode in Text-to-SQL systems.

## 10 LIMITATIONS

Our analysis, while systematic, has several limitations. First, we conduct all experiments on BIRD-mini for cost reasons. Second, we evaluate only closed-source frontier LLMs, which could restrict reproducibility. Third, we rely on LLM-generated sub-questions and this step may introduce subtle artifacts that influence model behavior. Additionally, our hallucination taxonomy and annotations use deterministic rule-based heuristics without large-scale human validation. Finally, the mitigation implications we propose are diagnostic insights derived from observed patterns, not fully developed or experimentally verified solutions.

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

## A APPENDIX: SAMPLE SPIDER BENCHMARK RESULTS

```
[
    {
        "path": [
            {
                "question_id": 786,
                "db_id": "world_1",
```

```
648            "question": "Which unique cities are in Asian countries where Chinese is an
649                ↪ official language?",
650            "SQL": "select distinct t3.name from country as t1 join countrylanguage as t2
651                ↪ on t1.code = t2.countrycode join city as t3 on t1.code = t3.
652                ↪ countrycode where t2.isofficial = 't' and t2.language = 'chinese' and
653                ↪ t1.continent = 'asia'",
654            "predicted_query": "SELECT DISTINCT city.Name FROM city JOIN country ON city.
655                ↪ CountryCode = country.Code JOIN countrylanguage ON country.Code =
656                ↪ countrylanguage.CountryCode WHERE country.Continent = 'Asia' AND
657                ↪ countrylanguage.Language = 'Chinese' AND countrylanguage.IsOfficial =
658                ↪ 'T'",
659            "evidence": "",
660            "correct": 1,
661            "error": null,
662            "is_original": true
663          }
664        ]
665      },
666      {
667        "path": [
668          {
669            "question_id": 896,
670            "db_id": "network_1",
671            "question": "\"Show names of all high school students who are not friends with
672                ↪  anyone.\"",
673            "SQL": "SELECT name FROM Highschooler EXCEPT SELECT T2.name FROM Friend AS T1
674                ↪ JOIN Highschooler AS T2 ON T1.student_id = T2.id",
675            "predicted_query": "SELECT 'name' FROM 'Highschooler' WHERE 'ID' NOT IN (
676                ↪ SELECT 'student_id' FROM 'Friend' ) AND 'ID' NOT IN ( SELECT '
677                ↪ friend_id' FROM 'Friend' )",
678            "evidence": "",
679            "correct": 0,
680            "error": null,
681            "is_original": true,
682             "hallucination": [
683                [
684                    "Logic-Based: Clause Abuse",
685                    "Unexpected clause 'SELECT'"
686                ],
687                [
688                    "Logic-Based: Clause Abuse",
689                    "Unexpected clause 'FROM'"
690                ],
691                [
692                    "Logic-Based: Clause Abuse",
693                    "Unexpected clause 'WHERE'"
694                ],
695            ]
696          }
697        ]
698      },
699      {
700        "path": [
701          {
702            "question_id": 1033,
703            "db_id": "real_estate_properties",
704            "question": "What are the names of properties?",
705            "SQL": "SELECT property_name FROM Properties",
706            "predicted_query": "SELECT 'property_name' FROM 'Properties'",
707            "evidence": "",
708            "correct": 1,
709            "error": null,
710            "is_original": false
711          },
712          {
713            "question_id": 1033,
714            "db_id": "real_estate_properties",
715            "question": "What are the names of properties that are houses?",
716            "SQL": "SELECT property_name FROM Properties WHERE property_type_code = 'House
717                ↪ '",
718            "predicted_query": "SELECT 'property_name' FROM 'Properties' JOIN '
719                ↪ Ref_Property_Types' ON 'Properties'.'property_type_code' = '
720                ↪ Ref_Property_Types'.'property_type_code' WHERE '
721                ↪ property_type_description' = 'House'",
722            "evidence": "",
723            "correct": 0,
724            "error": null,
725            "is_original": false,
726            "hallucination": [
727                [
728                    "Schema-Based: Attribute Overanalysis",
729                    "Extra column 'property_type_description'",
```

```
702               ],
703               [
704                   "Logic-Based: Clause Abuse",
705                   "Unexpected clause 'ON'"
706               ],
707               [
                      "Logic-Based: Join Redundancy",
708                   "1 extra JOIN(s)"
709               ]
710           ]
711       },
712       {
            "question_id": 1033,
713         "db_id": "real_estate_properties",
            "question": "What are the names of properties that are either houses or
714           ↪ apartments with room count greater than 1?",
715         "SQL": "SELECT property_name FROM Properties WHERE property_type_code = 'House
              ↪ ' UNION SELECT property_name FROM Properties WHERE property_type_code
716           ↪ = 'Apartment' AND room_count > 1",
717         "predicted_query": "SELECT 'property_name' FROM 'Properties' p JOIN '
              ↪ Ref_Property_Types' pt ON p.'property_type_code' = pt.'
718           ↪ property_type_code' WHERE pt.'property_type_description' IN ('house',
719           ↪ 'apartment') AND p.'room_count' > 1",
            "evidence": "",
720         "correct": 0,
            "error": null,
721         "is_original": true,
            "hallucination": [
722             [
                    "Schema-Based: Attribute Overanalysis",
723                 "Extra column 'property_type_description'",
724             ],
                [
725                 "Logic-Based: Clause Abuse",
                    "Unexpected clause 'ON'"
726             ],
                [
727                 "Logic-Based: Join Redundancy",
                    "1 extra JOIN(s)"
728             ]
729         ]
730       }
      ]
731   },
732   {
        "path": [
733         {
                "question_id": 599,
734             "db_id": "tvshow",
                "question": "\"What are the contents of TV channels?\"",
735             "SQL": "SELECT Content FROM TV_Channel",
                "predicted_query": "SELECT 'Content' FROM 'TV_Channel'",
736             "evidence": "",
                "correct": 1,
737             "error": null,
                "is_original": false
738         },
            {
739             "question_id": 599,
                "db_id": "tvshow",
740             "question": "What is the content of the series named 'Sky Radio'?",
                "SQL": "SELECT Content FROM TV_Channel WHERE series_name = 'Sky Radio' ;",
741             "predicted_query": "SELECT 'Content' FROM 'TV_Channel' WHERE 'series_name' = '
                  ↪ Sky Radio'",
742             "evidence": "",
                "correct": 1,
743             "error": null,
                "is_original": false
744         },
            {
745             "question_id": 599,
                "db_id": "tvshow",
746             "question": "What is the content of the TV series named 'Sky Radio'?",
                "SQL": "SELECT Content FROM TV_Channel WHERE series_name = 'Sky Radio';'",
747             "predicted_query": "SELECT 'Content' FROM 'TV_Channel' WHERE 'series_name' = '
                  ↪ Sky Radio'",
748             "evidence": "",
                "correct": 1,
749             "error": null,
                "is_original": true
750         }
```

```
756              ]
757          },
758          {
759              "path": [
760                  {
761                      "question_id": 212,
762                      "db_id": "flight_2",
763                      "question": "How many flights fly from a specific source airport to a specific
                          ↪   destination airport?",
764                      "SQL": "SELECT count() FROM FLIGHTS AS T1 JOIN AIRPORTS AS T2 ON T1.
                          ↪   DestAirport = T2.AirportCode JOIN AIRPORTS AS T3 ON T1.SourceAirport =
                          ↪   T3.AirportCode",
765                      "predicted_query": "SELECT COUNT() FROM 'flights' WHERE 'SourceAirport' = '
                          ↪   SpecificSourceCode' AND 'DestAirport' = 'SpecificDestCode'",
766                      "evidence": "",
767                      "correct": 0,
768                      "error": null,
769                      "is_original": false,
770                      "hallucination": [
771                          [
772                              "Logic-Based: Clause Abuse",
773                              "Unexpected clause    SELECT    "
774                          ],
775                          [
776                              "Logic-Based: Clause Abuse",
777                              "Unexpected clause    FROM    "
778                          ],
779                          [
780                              "Logic-Based: Clause Abuse",
781                              "Unexpected clause 'WHERE'"
782                          ]
783                      ]
784                  },
785                  {
786                      "question_id": 212,
787                      "db_id": "flight_2",
788                      "question": "How many flights fly from Aberdeen to Ashley?",
789                      "SQL": "SELECT count() FROM FLIGHTS AS T1 JOIN AIRPORTS AS T2 ON T1.
                          ↪   DestAirport = T2.AirportCode JOIN AIRPORTS AS T3 ON T1.SourceAirport =
                          ↪   T3.AirportCode WHERE T2.City = 'Ashley'",
790                      "predicted_query": "SELECT COUNT() FROM 'flights' WHERE 'SourceAirport' = (
                          ↪   SELECT 'AirportCode' FROM 'airports' WHERE 'City' = 'Aberdeen') AND '
                          ↪   DestAirport' = (SELECT 'AirportCode' FROM 'airports' WHERE 'City' = '
                          ↪   Ashley')",
791                      "evidence": "",
792                      "correct": 0,
793                      "error": null,
794                      "is_original": false,
795                      "hallucination": [
796                          [
797                              "Logic-Based: Clause Abuse",
798                              "Unexpected clause    SELECT    "
799                          ],
800                          [
801                              "Logic-Based: Clause Abuse",
802                              "Unexpected clause    FROM    "
803                          ],
804                          [
805                              "Logic-Based: Clause Abuse",
806                              "Unexpected clause 'WHERE'"
807                          ]
808                      ]
809                  },
                  {
                      "question_id": 212,
                      "db_id": "flight_2",
                      "question": "How many flights fly from Aberdeen to Ashley?",
                      "SQL": "SELECT count() FROM FLIGHTS AS T1 JOIN AIRPORTS AS T2 ON T1.
                          ↪   DestAirport = T2.AirportCode JOIN AIRPORTS AS T3 ON T1.SourceAirport =
                          ↪   T3.AirportCode WHERE T2.City = 'Ashley' AND T3.City = 'Aberdeen'",
                      "predicted_query": "SELECT COUNT() FROM 'flights' WHERE 'SourceAirport' = (
                          ↪   SELECT 'AirportCode' FROM 'airports' WHERE 'City' = 'Aberdeen') AND '
                          ↪   DestAirport' = (SELECT 'AirportCode' FROM 'airports' WHERE 'City' = '
                          ↪   Ashley')",
                      "evidence": "",
                      "correct": 0,
                      "error": null,
                      "is_original": true,
                      "hallucination": [
                          [
                              "Logic-Based: Clause Abuse",
```

Table 1: Taxonomy of hallucination types observed in failed SQL generations, originally adopted from Qu et al. (2024).

| Category | Description |
|---|---|
| Schema-Based: Schema Contradiction | The predicted query uses invalid or unknown tables, columns, or aliases not present in the database schema. Also includes misuse of wildcard or backtick syntax. |
| Schema-Based: Attribute Over-analysis | The query introduces valid but unnecessary tables or columns that are not present in the ground truth, resulting in over-specific or redundant retrieval logic. |
| Schema-Based: Value Misrepresentation | The query mishandles data representation, such as incorrect or missing type casts, or inconsistent literal values. |
| Logic-Based: Join Redundancy | The query contains more JOIN operations than the ground truth, indicating hallucinated or spurious table relationships. |
| Logic-Based: Clause Abuse | The query includes structural SQL clauses (e.g., `GROUP BY`, `LIMIT`, `ORDER BY`) or logical operators (e.g., `AND`, `OR`) that were absent in the ground truth. |
| Logic-Based: Mathematical Delusion | The query exhibits invalid or misleading numerical reasoning, such as uncasted division, misuse of `%`, improper use of `BETWEEN`, or syntax errors in arithmetic expressions. |

Table 2: Recurrent vs. emergent hallucination definitions in the drill-down analysis.

| Category | Description |
|---|---|
| Recurrent Hallucination | A hallucination that occurs somewhere in the drill-down path and reappears again in the final step. |
| Emergent Hallucination | A hallucination that manifests in the final step of the drill-down path with no prior instances in earlier steps. |

```
                    "Unexpected clause    SELECT   "
              ],
              [

                  "Logic-Based: Clause Abuse",
                  "Unexpected clause    FROM   "
              ],
              [

                  "Logic-Based: Clause Abuse",
                  "Unexpected clause 'WHERE'"
              ]
          ]
      }
    ]
  }
]
```

# B    APPENDIX: TAXONOMY OF HALLUCINATION TYPES

# C    APPENDIX: EXPERIMENT RESULTS (EX, SOFT-F1)

# D    APPENDIX: HALLUCINATION ANNOTATION RULES

This appendix details the deterministic rule-based procedure we use to annotate hallucination categories in predicted SQL queries. Each predicted query is compared against its ground-truth SQL and the corresponding BIRD schema. The rules below correspond exactly to the implementation used during evaluation.

Table 3: BIRD-mini EX Accuracy (%) and Soft F1-Scores across Difficulty Levels

| Model | Simple | Moderate | Challenging | Total |
|---|---|---|---|---|
| *Count* | 148 | 250 | 102 | 500 |
| Claude-3.5-Sonnet | 56.08 | 35.20 | 23.53 | 39.00 (EX) |
|  | 59.39 | 38.61 | 31.15 | 43.24 (F1) |
| Claude-3.5-Sonnet (+ History) | 50.00 | 34.40 | 20.59 | 36.20 (EX) |
|  | 56.93 | 37.84 | 29.01 | 41.69 (F1) |
| Claude-3.7-Sonnet | 51.35 | 38.40 | 21.57 | 38.80 (EX) |
|  | 55.09 | 43.58 | 29.59 | 44.13 (F1) |
| Claude-3.7-Sonnet (+ History) | 52.70 | 38.40 | 24.51 | 39.80 (EX) |
|  | 57.32 | 42.10 | 30.78 | 44.30 (F1) |
| GPT-4.0-Turbo | 58.78 | 34.00 | 17.65 | 38.00 (EX) |
|  | 60.66 | 38.45 | 24.14 | 42.11 (F1) |
| GPT-4.0-Turbo (+ History) | 55.41 | 37.60 | 19.61 | 39.20 (EX) |
|  | 57.45 | 40.48 | 25.16 | 42.38 (F1) |
| GPT-4.0-o-Mini | 47.97 | 31.60 | 13.73 | 32.80 (EX) |
|  | 50.87 | 34.42 | 20.63 | 36.48 (F1) |
| GPT-4.0-o-Mini (+ History) | 47.30 | 31.20 | 14.71 | 32.60 (EX) |
|  | 48.82 | 33.57 | 20.10 | 35.34 (F1) |
| GPT-4.1-nano | 47.97 | 26.80 | 11.76 | 30.00 (EX) |
|  | 50.05 | 29.40 | 19.49 | 33.49 (F1) |
| GPT-4.1-nano (+ History) | 50.68 | 28.40 | 15.69 | 32.40 (EX) |
|  | 52.50 | 32.16 | 18.89 | 35.47 (F1) |
| GPT-4.1-Mini | 59.46 | 41.60 | 21.57 | 42.80 (EX) |
|  | 61.47 | 45.04 | 28.48 | 46.53 (F1) |
| GPT-4.1-Mini (+ History) | 56.08 | 39.20 | 18.63 | 40.00 (EX) |
|  | 58.33 | 42.33 | 24.09 | 43.35 (F1) |

## D.1 ANNOTATION PIPELINE OVERVIEW

For each step in a drill-down path, we execute the following procedure:

1. **Parse SQL structure:** Extract tables, columns, aliases, alias–column pairs, SELECT elements, JOIN structures, and literal values from both the ground-truth and predicted queries.

2. **Normalize queries:** Convert both SQL strings to lowercase and uppercase variants to support rule-specific pattern matching.

3. **Compare predicted SQL to the schema and ground truth:** Validate every table, column, alias, clause, and operator against:

   (a) the database schema,

   (b) the ground-truth SQL query, and

   (c) SQL structural constraints.

4. **Assign hallucination categories:** Violations are mapped to one of six main hallucination types.

5. **Deduplicate:** Each hallucination type is counted at most once per query step, though multiple subcategories may be recorded.

We adopt the hallucination taxonomy described in the main paper: Schema-Based (Schema Contradiction, Attribute Overanalysis, Value Misrepresentation) and Logic-Based (Clause Abuse, Mathematical Delusion, Join Redundancy).

## D.2 SCHEMA-BASED HALLUCINATIONS

### D.2.1 SCHEMA CONTRADICTION

A prediction is labeled as a **Schema Contradiction** when it violates the database schema in any of the following ways:

- **Unknown tables:** Use of tables not present in the database schema.
- **Unknown columns:** Use of columns that do not appear in any table referenced in the query.
- **Alias errors:**
  - alias refers to a nonexistent table,
  - aliased column does not exist in the referenced table.
- **Wildcard or backtick misuse:** Presence of patterns such as `table.*` or MySQL-style backticks.
- **Missing table reference:** A column exists in the schema but is used without including its table in the FROM or JOIN clauses.

### D.2.2 ATTRIBUTE OVERANALYSIS

A prediction is labeled as **Attribute Overanalysis** when it adds valid but unnecessary schema elements that do not appear in the ground truth:

- **Extra columns** not used in the ground-truth query.
- **Extra tables** joined despite not being needed to answer the question.

This captures over-specification rather than invalid schema references.

### D.2.3 VALUE MISREPRESENTATION

A prediction is labeled as **Value Misrepresentation** when it mishandles literal values or type-casting semantics, such as:

- mismatched or altered literal values in the WHERE clause,
- unnecessary casts present in the prediction but absent in the ground truth,
- missing casts that appear in the ground truth.

## D.3 LOGIC-BASED HALLUCINATIONS

### D.3.1 CLAUSE ABUSE

A prediction is labeled as **Clause Abuse** when it introduces structural SQL clauses or logical constructs that do not appear in the ground truth, including:

- extra top-level clauses such as `GROUP BY`, `HAVING`, `ORDER BY`, `LIMIT`, or `OFFSET`,
- unnecessary logical operators such as `OR` or extraneous uses of `AND`,
- introduction of JOIN variants, CTEs, set operations, or vendor-specific syntax not present in the ground truth.

### D.3.2 MATHEMATICAL DELUSION

A prediction is labeled as **Mathematical Delusion** when it introduces faulty mathematical or operator semantics, such as:

- integer division without casting,
- misuse of the modulo operator (`%`) as a percentage,
- inappropriate use of `BETWEEN` outside valid numeric/date ranges,
- malformed or incomplete operator structures identified through SQL parser errors.

### D.3.3 JOIN REDUNDANCY

A prediction is labeled as **Join Redundancy** when it contains more JOIN operations than the ground truth, indicating hallucinated relational reasoning.

### D.4 DEDUPLICATION PROCEDURE

For each query step:

- Each *main* hallucination type is counted at most once.
- All *subcategory descriptions* are recorded for fine-grained analysis.
- An annotated list of hallucinations is stored in the evaluation JSON for reproducibility.

This rule-based annotation pipeline ensures that hallucination labels are precise, interpretable, and fully consistent across all models and all drill-down steps analyzed in the paper.

## E APPENDIX: PROMPTS

**Prompt 1: Progressive Question Rewriting Prompt**

```
Let's take this step-by-step.

Given this database schema: {schema_prompt}

Given this original question: "{original_question}"

Generate a new natural language question that maintains the same
structure and semantics but aligns with the following SQL query:

{partial_sql}

Do not include any information in your generated question that is
not directly included in the query. The original question should
be used as reference to generate this question.

Requirements: - The generated question must correspond exactly to
what this SQL retrieves - Maintain the same domain context and
terminology as the original question - The question should be
answerable using only this SQL query

Generate only the natural language question.
```

**Prompt 2: Text-to-SQL Prompt**

```
Using valid {sql_dialect} and understanding External Knowledge:
{knowledge}

{base_prompt}{knowledge_text}, answer the following questions for
the tables provided above. Generate the {sql_dialect} for the
above question after thinking step by step:

In your response, you do not need to mention your intermediate
steps.

Do not include any comments in your response.

Do not need to start with the symbol ''

You only need to return the result {sql_dialect} SQL code

start from SELECT
```

