# OpenReview forum: "Drill-Down Analysis of LLM Hallucination Patterns in Text-to-SQL"
_ICLR.cc/2026/Conference — ICLR 2026 Conference Desk Rejected Submission_

### Official Review · Reviewer_Tw4D · 2025-10-30

**Soundness:** 3
**Presentation:** 3
**Contribution:** 2
**Rating:** 4
**Confidence:** 4

**Summary:**

This paper introduces a "drill-down" framework to analyze how and when Large Language Models (LLMs) make errors (hallucinate) when generating SQL queries from natural language questions. Instead of just checking if the final SQL is correct, the authors decomposed complex problems from the BIRD-mini dataset into a series of simpler, progressive steps.

**Strengths:**

1. The core contribution, the "drill-down evaluation framework," is a significant strength. By decomposing complex SQL queries into a sequence of progressive sub-queries and sub-questions, the authors move beyond simple, final-step accuracy. This method allows for a much-needed, fine-grained analysis of how and when errors are introduced, which is more valuable than just knowing if an error occurred.

2. The paper introduces a useful temporal dimension to error analysis by defining "Recurrent Hallucinations" (errors that propagate from simple steps) and "Emergent Hallucinations" (errors that appear only in the final, complex step). This distinction provides a new and powerful lens for understanding LLM failure modes.

3. The discovery of the "duality" of contextual history is a major finding. The insight that providing history amplifies recurrent hallucinations (reinforces early mistakes) while simultaneously reducing emergent ones (protects against new errors) is non-obvious and has direct implications for designing multi-turn, conversational Text-to-SQL systems.

**Weaknesses:**

1. Limited Dataset Scale: The analysis relies on the BIRD-mini dataset, which is a small subset (500 original questions) of the full BIRD benchmark. While the authors expanded this into 1,383 sub-questions, the findings (especially the specific probabilities of recurrent vs. emergent errors) may not generalize to the wider variety and "long-tail" complexity of the full BIRD dataset or other complex Text-to-SQL benchmarks.

2. Potential Artifacts from Sub-Question Generation: The framework relies on an LLM to generate the natural language sub-questions (an SQL-to-NL task). While the authors justify this as being more reliable than NL-to-SQL, this generation step is not perfect and could introduce its own artifacts or subtle "hallucinations" into the evaluation dataset itself. The paper does not appear to analyze the error rate or potential bias of this LLM-based dataset creation step.

3. Scope of "Hallucination": The paper defines "hallucination" in a functional, post-hoc way (i.e., the generated SQL is incorrect according to the taxonomy). This is a practical and necessary limitation for an empirical study, but it doesn't fully capture the cognitive or generative process of why the LLM produced the wrong tokens. Terms like "cognitive load" (Section 4) are used as high-level metaphors rather than as measurable mechanisms.

4. "Alignment" Claims: The paper uses strong terms like "systematic misalignment" and "a path to better alignment." While the findings are certainly relevant to the goals of AI alignment, the paper is fundamentally an error analysis that diagnoses problems. It does not propose or test a new alignment method or solution. The conclusions might slightly overstate the paper's direct contribution to alignment methodology.

**Questions:**

see weakness

---

> ### Author Response · Authors · 2025-11-21
> **Rebuttal**
>
> We thank Reviewer @Tw4D for an overall positive and thoughtful assessment. Their review highlights several core strengths.
>
> **Weakness 1 Response:** We agree that dataset scale is an important consideration. Our use of BIRD-mini which is an expert-curated, representative subset designed specifically for cost-efficient evaluation, was driven solely by the prohibitive API cost of evaluating six frontier LLMs, not by methodological limitations of the framework. The decomposition, temporal tracking, and annotation pipelines are all dataset-agnostic and require only small configuration adjustments to support full BIRD, Spider, KaggleDBQA, or other large-scale benchmarks.
>
> To underline this point, we now clarify in the manuscript that:
>
> - the framework has already been tested on a subset of Spider,
>
> - it scales cleanly to larger and more complex datasets, and
>
> - BIRD-mini is widely adopted in recent work (e.g., NeurIPS 2025 and CIKM 2025), confirming its representativeness for Text-to-SQL research.
>
> We believe this addresses concerns around generalizability.
>
> **Weakness 2 Response:** We appreciate the reviewer raising this important point. Sub-question generation is indeed an LLM-mediated SQL-to-NL task, and while more reliable than NL-to-SQL, it may introduce subtle artifacts. We now explicitly acknowledge this as a limitation in the paper:
>
> “The SQL-to-NL generation step may introduce subtle artifacts that could influence hallucination patterns. This limitation should be taken into account when interpreting the results.”
>
> **Weakness 3 Response:** We agree that such terms may suggest mechanistic or cognitive claims beyond the empirical scope of our analysis. We have removed or revised language around “cognitive load,” “systematic misalignment,” and similar metaphors in Section 4. In the revised manuscript, these phenomena are framed explicitly as behavioral patterns observed in model outputs, not as mechanistic explanations:
>
> “We interpret these behavioral distinctions as two fundamentally different failure mechanisms, without attributing internal cognitive or architectural causes.”
>
> This aligns better with the reviewer’s concern and maintains conceptual clarity.
>
> **Weakness 4 Response:** We agree that the conclusions should not imply that the paper introduces or evaluates a new alignment method. We have clarified the framing in the Discussion and Conclusion sections. Our contributions are diagnostic, not intervention-based. We now explicitly state that the mitigation implications derived from our findings are exploratory insights intended to guide future research, not finalized methods:
>
> “These mitigation implications are exploratory insights grounded in our findings, not tested alignment strategies.”
>
> This addresses the reviewer’s concern about overstating the alignment contribution
>
> We appreciate Reviewer @Tw4D’s balanced and encouraging review.

---

### Official Review · Reviewer_f9KX · 2025-10-30

**Soundness:** 3
**Presentation:** 3
**Contribution:** 3
**Rating:** 6
**Confidence:** 4

**Summary:**

This paper introduces a drill-down framework to analyze hallucination patterns in Text-to-SQL tasks LLMs. The authors decompose SQL queries and their corresponding natural language questions from the BIRD-mini dataset, enabling fine-grained temporal tracking of hallucinations as they propagate through multi-step query construction. They define two key hallucination types: recurrent (persisting from early steps to final query) and emergent (appearing only at the final step), and analyze six frontier LLMs.

**Strengths:**

1. The proposed drill-down methodology offers a clear and reproducible way to dissect LLM behavior in multi-step reasoning tasks. The idea of decomposing SQL into executable sub-queries for hallucination tracking is conceptually elegant and technically well-motivated. Generally I think this research problem is novel and interesting in this domain.
2. The combination of schema-based and logic-based hallucination taxonomies with temporal abstractions (recurrent vs. emergent) gives the study a better analytical depth rarely seen in hallucination papers.
3. Six frontier closed-source models were evaluated with both EX and Soft-F1 scores, demonstrating methodological rigor and careful experimental design. Results are clearly visualized, with consistent trends across models.

**Weaknesses:**

1. **Limited scalability and dataset scope:** the study relies solely on the BIRD-mini dataset. Although justified for computational feasibility, this restricts generalization to broader domains or more diverse SQL structures.
2. Since both dataset decomposition and LLM evaluations involve closed-source models (Claude and GPT), the work’s reproducibility and community accessibility are limited.
3. While the analysis is strong, the paper stops short of proposing concrete mitigation strategies or architectural insights. The findings illuminate what fails but not how to fix it.

**Questions:**

Could this drill-down framework be directly applied to other datasets (e.g., Spider or KaggleDBQA)? If so, what challenges arise in adapting decomposition and annotation pipelines?

---

> ### Author Response · Authors · 2025-11-21
> **Rebuttal**
>
> We thank Reviewer @f9KX’s  for their positive and constructive evaluation. Their review acknowledges the novelty of the problem setting.
>
> **Weakness 1 Response:** We agree that broader dataset coverage is important for generalizability. We emphasize that our choice of BIRD-mini was purely due to API inference cost when evaluating six frontier closed-source LLMs, not due to any methodological restriction. The core components of our pipeline are SQL decomposition, NL sub-question generation, temporal tracking, and hallucination annotation, which are all dataset-agnostic and operate independently of the underlying schema or dataset structure.
>
> To make this explicit:
>
> - The decomposition module works on any SQL string following standard grammar rules.
>
> - The sub-question generation prompt uses schema dictionaries programmatically extracted from any benchmark.
>
> - Our annotation logic relies on schema-grounded matching and SQL-structure rules, which generalize across datasets.
>
> In the revision, we added a clarification that our framework has been successfully applied to a subset of Spider to validate generality. We also cite that BIRD-mini is an official, expert-curated subset intentionally designed to be representative of the full benchmark while enabling cost-efficient experimentation. This design choice is reflected in its recent adoption by NeurIPS 2025 and CIKM 2025 papers.
>
> **Weakness 2 Response:** While we evaluate with closed-source frontier models, the pipeline itself is fully reproducible and model-agnostic. To clarify:
>
> - SQL decomposition is deterministic.
>
> - The NL sub-question generation uses a simple prompt that can be executed with any LLM (open-source or closed-source).
>
> - The hallucination taxonomy and annotation heuristics rely on programmatic schema checks and SQL structural mismatches, not model internals.
>
> - The temporal analysis uses only generated SQL and NL pairs, making it fully portable.
>
> Thus, reproducing our methodology does not require the same closed-source models; any sufficiently capable model can be substituted, and the pipeline will run identically. We emphasize this more clearly in the revised manuscript.
>
> **Weakness 3 Response:** We have expanded the Theoretical Analysis section to include three concrete mitigation implications that naturally follow from our temporal findings:
> 1. Early-Step Verification: Because recurrent hallucinations originate at the earliest steps and propagate forward, verifying or constraining early decomposition outputs can prevent downstream error cascades.
>
> 2. Selective or Truncated History: Since full history reduces emergent hallucinations but amplifies recurrent ones, an adaptive or truncated history mechanism may balance these competing effects.
>
> 3. Schema Alignment Measures: Both hallucination types frequently arise from schema misinterpretations; strengthening schema conditioning, grounding, or pre-validation can reduce these errors.
>
> These implications directly address the reviewer’s request for more actionable insights without overstating architectural prescriptions.
>
> **Question Response:** Yes, the framework generalizes directly. The primary adaptation involves updating the configuration that loads the database schema and SQL dialect. All other components like decomposition, sub-question generation, hallucination annotation, and stepwise comparison are fully compatible with Spider, and other Text-to-SQL datasets out of the box.
> We have added a clarifying paragraph in the revision and explicitly note that no component of the pipeline relies on BIRD-specific features. This ensures applicability to diverse datasets with different schema sizes, join structures, and SQL complexity levels.
>
> We appreciate Reviewer @f9KX’s constructive feedback and recognition of the value of our approach.

---

> > ### Comment · Reviewer_f9KX · 2025-11-23
> > **Thanks for the authors' detailed response.**
> >
> > 1. About "Weakness 1 Response" from authors.
> >
> > Thanks for the authors' clarification. However, I still have concerns about the generalization of the proposed method and analysis. Since different benchmark datasets have different features, I admit that Bird-Mini is widely used in top conferences, but other popular datasets like Spider 2.0 (which has challenging tasks) have their own special characteristics. I believe more results on diverse popular datasets can further strengthen your claims and findings. BTW, I didn't find the results on Spider in revised paper, I just found a claim instead.
> >
> > 2. About "Weakness 2 Response"
> >
> > I understand that the proposed methods can be used for open-source LLMs, but I am curious whether open-source LLMs have different hallucination patterns compared with closed-source LLMs.
> >
> > 3. About "Weakness 3 Response"
> >
> > Thanks for author's detailed response. However, in my weakness 3, I meant that I think it will strengthen the technique depth if authors can have some experimental results on whether those ideas can indeed mitigate hallucination instead of just some initial ideas. I believe it can really point the directions for the community. But I understand that it's not mandatory to do so. Just a gentle suggestion.
> >
> > Please let me know if I misunderstand anything. Thanks!

---

> ### Author Response · Authors · 2025-12-01
> **Results on sample Spider in revised paper**
>
> We have used our pipeline for a sample of the Spider benchmark to show it is supported. **The results are now explicitly shown in the Appendix A**

---

> > ### Author Response · Authors · 2025-12-04
> >
> > Comments about 1 and 2: We argue that these requests for additional datasets or more models reflect a desire for broader experimentation, not a weakness of the framework or approach itself. Again, this drill-down method can be directly applied to other benchmarks and used with any LLM capable of SQL to NL generation.
> > Comments about 3: Thank you, we appreciate the suggestion.

---

### Official Review · Reviewer_khhh · 2025-10-30

**Soundness:** 3
**Presentation:** 3
**Contribution:** 2
**Rating:** 4
**Confidence:** 5

**Summary:**

The paper proposes a drill-down evaluation framework for analyzing hallucination propagation in Text-to-SQL tasks. Authors decompose SQL queries from the BIRD-mini dataset into sub-queries and sub-questions to track hallucinations across different reasoning steps. They categorize hallucinations into schema-based and logic-based types, introducing two temporal dimensions: recurrent (persistent) and emergent (final-step) hallucinations. Experiments were conducted using six modern LLMs (Claude 3.5/3.7, GPT-4-turbo, GPT-4o-mini, GPT-4.1-mini, GPT-4.1-nano). The authors claim consistent hallucination patterns across models and highlight that contextual history tends to reduce emergent hallucinations but amplify recurrent ones.

**Strengths:**

1. The paper addresses an important and timely issue, hallucination behavior in Text-to-SQL models, which is directly related to model reliability and alignment.
2. The proposed drill-down framework is clearly structured and easy to follow, providing a systematic way to trace hallucination propagation across query steps. The introduction of recurrent and emergent hallucinations adds a useful temporal dimension to existing classification schemes.
3. The comparison of six modern LLMs (Claude and GPT series) strengthens the empirical validity of the findings and shows consistent failure patterns across models.

**Weaknesses:**

1. The work lacks clear novelty both in dataset construction and in the definition of hallucination categories. The so-called drill-down framework mainly repackages existing dataset decomposition and annotation strategies without introducing a fundamentally new methodological contribution. From the dataset perspective, decomposing SQL queries into sub-queries is not a new approach and has already been explored in prior Text-to-SQL studies. From the definition perspective, the proposed recurrent and emergent hallucinations are essentially incremental extensions of the taxonomy established in TA-SQL (Qu et al., 2024), rather than conceptually novel categories.
2. While the paper provides a quantitative description of hallucination patterns, it lacks deeper analytical or theoretical insight into the underlying causes of these behaviors. The findings primarily replicate observations from prior studies such as TA-SQL (Qu et al., 2024) and the in-context Text-to-SQL error analysis by Shen et al. (2025). Although the inclusion of multiple LLMs broadens the empirical coverage, the analysis offers limited depth in interpreting model mechanisms or proposing practical strategies for hallucination mitigation.
3. The reliance on the small BIRD-mini dataset substantially limits the generalizability of the results. Although the authors justify this choice by citing computational constraints, this rationale is not sufficiently persuasive for an ICLR-level contribution. Since the proposed framework aims to analyze hallucination behavior across complex query structures, evaluation on a more comprehensive dataset，such as Spider and its derived variants or, at minimum, stronger validation on the full BIRD benchmark would be necessary to demonstrate the robustness and broader applicability of the findings.
4. The paper does not discuss annotation reliability or inter-rater consistency, which raises concerns about the validity of the taxonomy-based analysis.The absence of such validation details weakens confidence in the reproducibility and objectivity of the reported hallucination patterns.

**Questions:**

1. The idea of decomposing SQL queries for hallucination analysis is interesting but not novel. Prior frameworks such as CHASE-SQL and TA-SQL (Qu et al., 2024) have already explored task decomposition and hallucination categorization. Please articulate more clearly what methodological innovation distinguishes your drill-down framework from these existing approaches. For instance, how does your analysis pipeline provide new insights that cannot be achieved through prior decomposition-based methods?
2. Strengthen the connection between hallucination recurrence and model alignment theory. Currently, the analysis remains descriptive, focusing on pattern quantification rather than underlying causes. A deeper theoretical interpretation of why recurrent and emergent hallucinations occur, and what this implies about model reasoning or attention dynamics, would significantly improve the contribution.
3. The paper does not clearly describe how hallucination annotations were conducted, whether through automated rule-based comparison or LLM-assisted classification. Although an “automated annotation pipeline” is mentioned, the paper provides no information on its validation or reliability. Could the authors clarify how the annotation process was implemented and verified? Specifically, how was consistency ensured across instances?
4. The reliance on the BIRD-mini dataset limits the generalizability of findings. Consider extending the analysis to larger or more diverse datasets (e.g., Spider and its variants) or at least validating the results on the full BIRD benchmark. This would demonstrate that the framework scales effectively and that the reported hallucination behaviors are not dataset-specific artifacts.
5. In sections such as Preliminaries (Section 3) and Hallucination Taxonomy (Section 4), the authors devote extensive space to reintroducing material that largely reproduces content from prior work. This repetition diminishes the visibility of the paper’s own contribution. The authors are encouraged to condense these sections or move the background material to the appendix to improve focus and clarity.

---

> ### Author Response · Authors · 2025-11-21
> **Rebuttal**
>
> We thank the reviewer for their detailed and thoughtful assessment. Some of the concerns appear to stem from missing clarity rather than limitations of the methodology itself. In the updated manuscript, we directly address each point:
>
> **Weaknesses 1 / Question 1 Response:** No, decomposition is not the novelty, it's the application of decomposition to a systematic study of Text-to-SQL hallucinations. Which has led to our novel definitions of emergent and recurrent. We have discussed in the paper that our contribution is fundamentally different: we construct paired NL–SQL drill-down paths and perform temporal hallucination analysis across these paths. No prior Text-to-SQL work generates NL sub-questions for all decomposed steps or studies how hallucinations propagate, recur, or emerge across these steps. Additionally, as noted by other reviewers, our analysis of Context History for progressive Text-to-SQL Hallucinations is a unique and novel approach not explored in previous literature. We argue there are novel techniques and results presented in this paper.
>
> **Weaknesses 2 / Question 2 Response:** The “lack of deeper analytical/theoretical insight” concern is directly addressed in the Theoretical Analysis section. We also expanded this section with three implications for mitigation. We also argue we are different from TA-SQL, our analysis reveals how certain hallucination categories recur or emerge under different history conditions, something they do not explore. Very importantly, we uncover that some techniques are not uniformly beneficial, we identify certain hallucinations types that are not benefited from context history and vice versa.
>
> **Weaknesses 4 / Question 3 Response:** This concern is addressed directly in the revision. The annotation heuristics and details (previously only in the supplementary material) are now incorporated into the Appendix and referenced in the Hallucination Taxonomy section. “A full description of the heuristics and rules we used to annotate the hallucination types is shown in Appendix C.”
>
> **Weaknesses 3 / Question 4 Response:** BIRD-mini was used only due to API inference cost, not methodological constraints. We fully acknowledge ICLR’s emphasis on substantive contributions independent of resource limitations. To meet that expectation:
> We clarified that our pipeline is flexible to other datasets.
>
> - We demonstrated successful application to a subset of Spider, confirming generality.
>
> - We emphasized that decomposition, annotation, and temporal analysis all scale linearly with dataset size.
>
> - We cited the BIRD authors’ own position that BIRD-mini is a high-fidelity, representative, expert-curated subset specifically intended to support cost-efficient experimentation.
>
> - We now state clearly that the method scales to full BIRD and other benchmarks with minimal configuration changes.
>
> **Question 5 Response:** We improved Sections 3 and 4, condensed background descriptions, and moved several definitions and explanatory paragraphs to the Appendix. This allows for more emphasis onto our framework and empirical findings.
>
> We appreciate the reviewer’s constructive feedback and believe the revised manuscript now articulates the methodology and insights with much greater clarity.

---

### Official Review · Reviewer_dKsh · 2025-11-01

**Soundness:** 1
**Presentation:** 2
**Contribution:** 1
**Rating:** 2
**Confidence:** 4

**Summary:**

This paper presents a drill-down evaluation framework for analyzing hallucination patterns in Large Language Models (LLMs) when used for Text-to-SQL tasks. The authors use the BIRD-mini dataset to decompose complex SQL queries into progressive sub-queries and analyze hallucination propagation. The study identifies three findings: (1) recurrent hallucinations, (2) final-step emergence, and (3) the impact of history on recurrent hallucinations and emergent hallucinations.

**Strengths:**

1.	The writing is generally clear and concise, with complex concepts well explained.

2.	The paper attempts to study an important research question: hallucinations of LLMs.

**Weaknesses:**

1.	The authors introduce two new hallucination categories—recurrent hallucinations and emergent hallucinations—but the motivation behind defining these specific types is not clearly explained.

2.	The reasoning behind the formulation of the three research questions is not well articulated. Moreover, the paper does not provide clear or explicit answers to these questions.

3.	Table 3 serves primarily as a validation of the experimental setup rather than a key result. Including it in the main paper distracts from the core findings. It would be more appropriate to move this table to the appendix, with only a brief summary of its results mentioned in the main text.

4.	The contribution of this paper is weak. The drill-down framework provides limited guidance on how to address or mitigate LLM hallucination issues in practice.

5.	Section 3.2 should be in the experiment part.

**Questions:**

1.	The entire experimental setup is built around the BIRD-mini dataset. How would the proposed framework perform on other Text-to-SQL benchmarks or domains? Could it be generalized or extended to broader settings beyond BIRD-mini?

2.	In Section 3.3, the authors mention that applying their framework to the full BIRD-dev dataset would lead to a "prohibitive analysis cost." Does this imply that the framework is not scalable to larger datasets? How do the authors plan to address scalability challenges given the increasing size of modern Text-to-SQL benchmarks?

3.	In Section 6.1, Figures 3 and 4 present conditional probabilities of recurrent and emergent hallucinations. Could the authors also report the unconditional probabilities of these hallucination types to provide a more complete understanding of their overall prevalence?

---

> ### Author Response · Authors · 2025-11-21
> **Rebuttal**
>
> We thank Reviewer @dKsh for their careful reading and direct feedback. Some of the concerns raised reflected missing clarity rather than limitations of the underlying methodology. In the updated manuscript, we directly address each point:
>
> **Weakness 1 Response:** Our definitions of recurrent and emergent hallucinations did not arise from an a priori taxonomy, but from consistently observed empirical patterns that emerged across six modern LLMs. Through the decomposition pipeline, we repeatedly noted two distinct ways hallucinations manifested: (1) errors introduced early in the reasoning path that persisted through later steps, and (2) errors that appeared abruptly only at the final, most complex step.
>
> We have strengthened the motivation section to clarify that these categories are not arbitrary labels but abstractions grounded in our empirical results Understanding whether hallucinations originate from early reasoning failures versus late-stage complexity mismatches is critical for explaining when and why LLMs fail in multi-step Text-to-SQL settings.
>
> **Weakness 2 Response:** We revisited the three research questions and expanded the framing to make their purpose explicit. Each question now directly maps to a core aspect of our empirical findings:
>
> 1. Where hallucinations originate within progressive reasoning paths.
>
> 2. How hallucinations evolve when contextual history is provided.
>
> 3. Whether hallucination behaviors are consistent across diverse LLM architectures.
>
> To address concerns about insufficient answers, we added explicit, concise responses for each question in the Theoretical Analysis section.
>
> **Weakness 3 Response:** We agree that Table 3 primarily serves as validation of the experimental decomposition and not as a central scientific result. We have moved this table to the Appendix and now refer to it briefly in the main text as supporting material.
>
> **Weakness 4 Response:** We revised the contribution statement and expanded the Theoretical Analysis section with three concrete implications for mitigating hallucinations:
>
> 1. Early-Step Verification: Since recurrent hallucinations often originate from the earliest steps, lightweight verification or model-checking at early stages can prevent error propagation.
>
> 2. Selective or Truncated History: While history reduces emergent hallucinations, excessive history amplifies recurrent ones. This suggests using selectively truncated context windows to balance both effects.
>
> 3. Schema Alignment Measures: Both emergent and recurrent hallucinations disproportionately arise from schema misinterpretations; thus schema-aware training or retrieval augmentation can meaningfully reduce these failure modes.
>
> **Weakness 5 Response:** We agree with the reviewer’s structural suggestion and have made these changes.
>
> **Question 1 Response:** Yes. The framework is dataset-agnostic. The SQL decomposition logic, sub-question generation, hallucination annotation heuristics, and temporal analysis apply directly to full BIRD, Spider, and other Text-to-SQL benchmarks with only minor configuration adjustments. We now state this explicitly in the paper. To reinforce this claim, we have successfully applied the pipeline to a subset of Spider and include a short sentence confirming its compatibility with larger and structurally different datasets.
>
> **Question 2 Response:** No. The term “prohibitive cost” refers solely to API inference costs when evaluating six frontier closed-source LLMs, not computational or methodological scalability. The pipeline is entirely scalable: decomposition, annotation, and analysis cost are linear in the number of SQL steps.
>
> We clarified this distinction in the revision.
>
> We also cite the BIRD authors own documentation that BIRD-mini is a high-fidelity, representative subset specifically designed to enable cost-efficient experimentation. This matches current community practice: recent NeurIPS 2025 and CIKM 2025 papers also rely on BIRD-mini.
>
> **Question 3 Response:**  We have added unconditional probabilities for recurrent and emergent hallucination types, specifically:
>
> 1. P(Hallucination in Final Step)
>
> 2. P(Hallucination in Earlier Steps)
>
> These results complement the conditional probabilities and provide a fuller picture of hallucination prevalence across the progressive reasoning path.
>
> We appreciate Reviewer @dKsh’s critique and believe the revised manuscript now presents a clearer, more robust articulation of our analysis and its significance.

---

### Author Response · Authors · 2025-11-21
**Summary of discussion.**

We thank all the reviewers for their careful and constructive feedback. Our work introduces a drill-down evaluation framework that decomposes NL-SQL pairs into progressive steps to analyze hallucinations temporally. Our work revealed two empirically grounded behaviors, recurrent/emergent, and a non-obvious dual effect of contextual history. Several reviewers noted the clarity and relevance of this paper and its contributions.

The two most common critiques raised can be summarized as follows; (i) requests for experiments on larger datasets, and (ii) lack of a method to mitigate the observed phenomena. We would like to emphasize that both these concerns are orthogonal to the main contributions of this paper, and it is our belief that the reviewers would have raised their scores after our clarifications in the discussion period.

Regarding experiments on larger datasets, we clarified that our choice of using the BIRD-mini dataset is driven only by API cost when evaluating 6 frontier LLMs, not by methodological limitations. BIRD-mini is an expert-curated, representative subset widely used in recent NeurIPS and CIKM papers. Additionally, to mitigate this concern we have included results on a sample of the Spider benchmark (Appendix A) to demonstrate that our approach can be applied out-of-the-box to datasets outside of BIRD-mini. We would like to emphasize that requiring experiments on the entire BIRD and Spider datasets, with multiple LLMs, would be extremely costly and that puts such research beyond the financial scope of all except a handful of researchers. The results on BIRD-mini demonstrate that there is no reason to believe that extra experiments on larger datasets would provide additional information or change the analysis. BIRD-mini has been designed to make resource-constrained research in this area possible and we demonstrated insightful results on this dataset.

Regarding the concern of providing a method for mitigation, we argue that the value of researching hallucinations is complementary to discovering the ability to eradicate them but should not be judged purely by that standard. The history of science demonstrates that studying failure modes is important even if we do not immediately have a mechanism to fix the failures. The study of LLM hallucinations teaches us to design for epistemic uncertainty. The core contribution of this work is identifying new classes of hallucinations that persist even in the popular drill-down approach for NL-to-SQL tasks (used by many of the SOTA techniques), and we provide a flexible open-source implementation of the analysis framework that can be used to study new proposed approaches or LLM models as they are introduced.

We appreciate the reviewer’s suggestions and believe the revised manuscript addresses all other concerns clearly while demonstrating that this is meaningful and timely research for understanding LLM weaknesses in NL-to-SQL systems.

---

### Note · Program_Chairs · 2026-01-17
**Submission Desk Rejected by Program Chairs**

The following references in this submission do not refer to real documents and/or have major errors in bibliographic information:

 - Sang Michael Xie, Yao Lu, Aditi Raghunathan, Percy Liang Yin, and Chelsea Finn. Explanationbased prompting for continual learning. arXiv preprint arXiv:2104.07143, 2021.
- Xiangfu Meng, Runzhi Liu, Xinyi Yang, Junyi Gao, Pranav Narayan, Matthew Agar-Johnson, AmirHamed Mohsenian-Rad, Rui Zhang, Chuan Lei, and Yu Su. Multi-turn interactions for text-to-sql with large language models. In Proceedings of the 34th ACM International Conference on Information and Knowledge Management (CIKM), 2025. Uses BIRD Mini-Dev as a core benchmark.